# Human adenylyl cyclase 9 is auto-stimulated by its isoform-specific C-terminal domain

Zhihao Chen, Ferenc A Antoni

**Human transmembrane adenylyl cyclase 9 (AC9) is not regulated by heterotrimeric G proteins. Key to the resistance to stimulation by Gs-coupled receptors (GsRs) is auto-inhibition by the COOH-terminal domain (C2b). The present study investigated the role of the C2b domain in the regulation of cyclic AMP production by AC9 in HEK293FT cells expressing the GloSensor22F cyclic AMP-reporter protein. Surprisingly, we found C2b to be essential for sustaining the basal output of cyclic AMP by AC9. A human mutation (E326D) in the parallel coiled-coil formed by the signalling helices of AC9 dramatically increased basal activity, which was also dependent on the C2b domain. Intriguingly, the same mutation enabled stimulation of AC9 by GsRs. In summary, auto-regulation by the C2b domain of AC9 sustains its basal activity and quenches activation by GsR. Thus, AC9 appears to be tailored to support constitutive activation of cyclic AMP effector systems. A switch from this paradigm to stimulation by GsRs may be occasioned by conformational changes at the coiled-coil or removal of the C2b domain.**

## Introduction

Transmembrane adenylyl cyclases produce the ubiquitous signalling molecule adenosine-3′:5′-monophosphate (cAMP). Nine genes encode these enzymes in mammals, and each paralogue has unique regulatory properties (Ostrom et al, 2022). Adenylyl cyclase 9 (AC9) is widely distributed in the body and has been implicated in a number of physiological processes, including cardiac function, body fat mass and body weight, and cancer pathologies and atherosclerosis (Antoni, 2020; Ostrom et al, 2022). Partial high-resolution maps of the structure of AC9 obtained by cryo-electron microscopy have been recently published (Qi et al, 2019, 2022). In brief, the 1,353-residue single polypeptide chain of AC9 forms a tripartite structure. This consists of a large transmembrane array that is connected to the catalytic domain in the cytoplasm by two α-helices (Fig 1A).

The helices form a short parallel coiled-coil in close proximity to the catalytic core (Fig 1A). Sequence alignments of human adenylyl cyclases indicate that the coiled-coil of the "signalling helices" (Anantharaman et al, 2006; Bassler et al, 2018) is a generic feature of these proteins (Fig 1B). Coiled-coils are important regulatory modules in several proteins (Lupas & Bassler, 2017) including soluble guanylyl cyclases where the binding of nitric oxide is transmitted to the catalytic domains through conformational changes of the signalling helix (Horst et al, 2019).

Significantly, the enzymatic activity of full-length AC9 is largely insensitive to heterotrimeric G proteins (Pálvölgyi et al, 2018; Baldwin et al, 2019; Qi et al, 2019). In the case of Gi/o, this is due to the lack of a suitable binding pocket (Baldwin et al, 2019). With respect to Gs, the isoform-specific carboxyl-terminal (C2b) domain exerts a seemingly paradoxical auto-inhibitory effect by occluding the active site in the presence of Gsα-GTP (Qi et al, 2019). Given the unique regulatory features of AC9, this study investigated further the role of the C2b domain and its potential interactions with the coiled-coil.

The effects of a previously reported missense mutation, E326D (Calebiro et al, 2016), on cAMP production by human AC9 were analysed. This mutation is at the interface of the coiled-coil (Fig 1C and D). The results showed the E326D mutation markedly increased basal cAMP production by AC9, which was largely dependent on the presence of the isoform-specific C2b domain. In parallel, the mutation reduced the efficacy of the C2b domain to quench the activation of AC9 by Gs-coupled receptors (GsRs). Finally, we show that the basal activity of WT AC9 also requires the C2b domain.

## Results and Discussion

### Effect of the coiled-coil mutation

Mutation E326D caused a 10-fold increase in basal cAMP production and an enhancement of the cAMP response to isoproterenol (see Fig 2A and statistical summary in Fig 3). Consistent with previous results with a different type of assay (Pálvölgyi et al, 2018), no isoproterenol-induced cAMP response attributable to WT AC9 could be reliably

Centre for Discovery Brain Sciences, Deanery of Biomedical Sciences, University of Edinburgh, Edinburgh, UK

Correspondence: ferenc.antoni@ed.ac.uk
Zhihao Chen's present address is RiboX Therapeutics, Shanghai, China

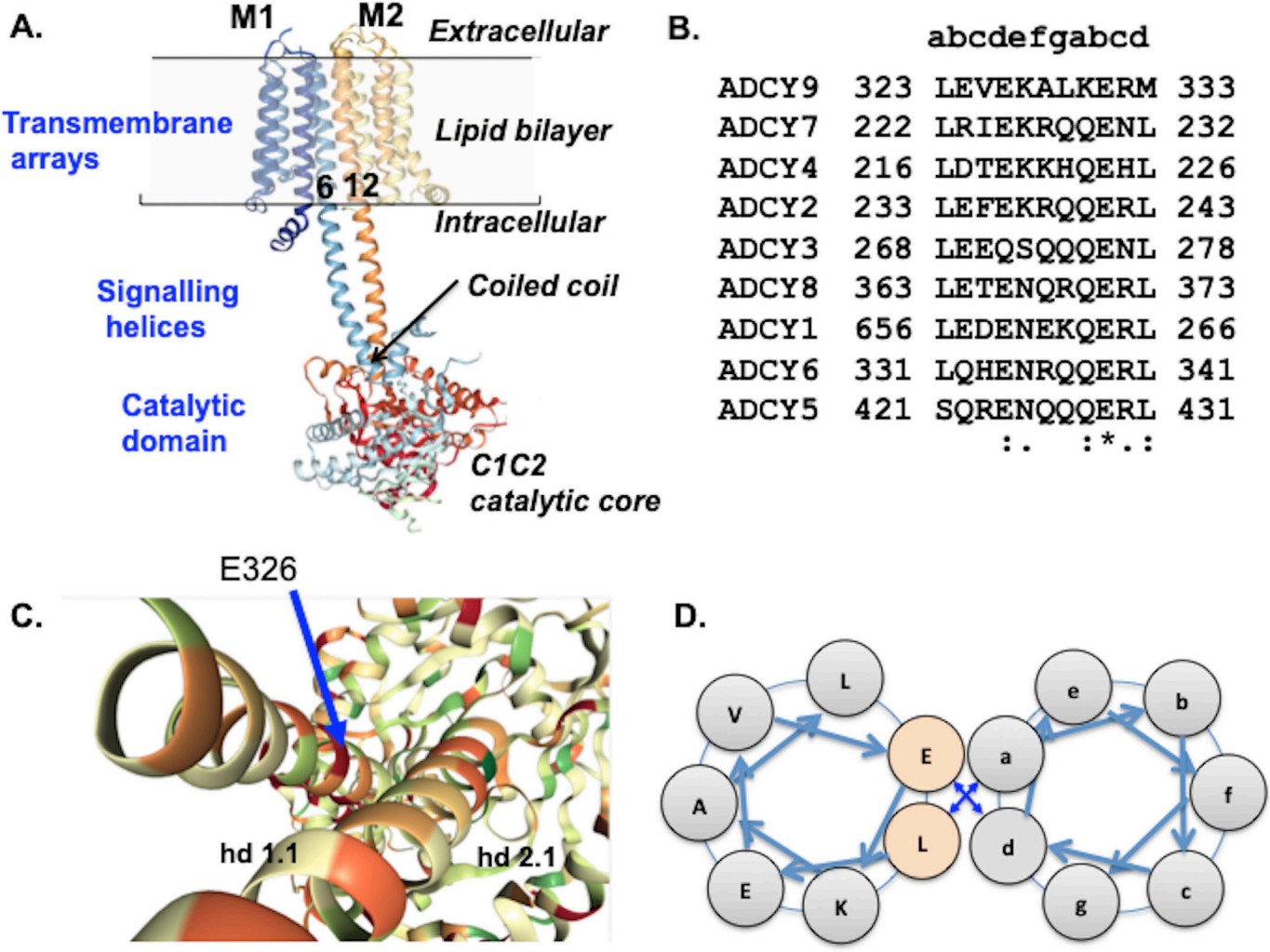

**Figure 1.  Considerations of protein structure guiding this study.**
**(A)** Tripartite structure of AC9 derived from cryo-EM studies by Qi and co-workers (Qi et al, 2019, 2022). Note that the enzyme consists of a 1,353-residue single polypeptide and that some parts of the protein (shown by a dashed red line) have not been resolved in the cryo-EM studies (Antoni, 2020). Transmembrane helices 6 and 12 are interfaced in the lipid bilayer and continue into the cytoplasm as the signalling helices that form a short, parallel coiled-coil just above the catalytic domain. **(B)** Alignment of the hd1.1 (Qi et al, 2019, 2022) segment of human adenylyl cyclase paralogues. The small lettering in the top row indicates the positions in the coiled–coiled heptad of bovine AC9. **(C, D)** E326 is at the interface of the coiled-coil, at position *d* of the heptad repeat as schematically shown in (D).

discerned. Others reported that a mutation in the predicted coil–coil of AC5 (M1029K) led to an enhanced GsR-induced cAMP response (Lupas & Bassler, 2017; Doyle et al, 2019; Qi et al, 2019); however, no changes in basal cAMP levels were found. In the case of AC9, aspartate instead of glutamate at the coiled-coil interface (E326D) in all probability changes the conformation of the coil (Jonson & Petersen, 2001; Straussman et al, 2007). The mutation led to a large, close to a 10-fold increase in basal cAMP levels. In parallel, the cAMP response to GsR activation appeared, indicating a release from the potent auto-inhibitory effect exerted by the C2b domain (Pálvölgyi et al, 2018; Qi et al, 2019). The presence of such an adenylyl cyclase in thyroid epithelial cells (Calebiro et al, 2016) is likely to lead to hypertrophy and hyperplasia. In the context of a second oncogenic mutation, it can support adenomatous hyperproliferation (Calebiro et al, 2016) or epithelial–mesenchymal transition–producing malignant tumour growth (Tan et al, 2018).

## Probing the role of the C2b domain

Given the prominent role of the C2b domain in the regulation of the activity of AC9 (Pálvölgyi et al, 2018; Qi et al, 2019), we examined the effects of its deletion on cAMP levels produced by AC9 and E326D_AC9. Surprisingly, removal of the C2b domain from E326D_AC9 (E326D_AC9C2a) and WT AC9 (AC9C2a) reduced basal cAMP levels by 80–90% (see Figs 2B and C and 3). With respect to stimulation by GsR, the amplitude of the agonist-induced cAMP response of AC9C2a was dramatically enhanced when compared to full-length AC9 (Fig 2B). This is fully consonant with previous results obtained by different methods of analysis (Pálvölgyi et al, 2018; Qi et al, 2019). However, it was not the case for E326D_AC9C2a: the peak levels of cAMP and the time course of the isoproterenol response were not consistently different from those of full-length E326D_AC9 at any concentration of agonist tested, indicating that the efficacy of auto-

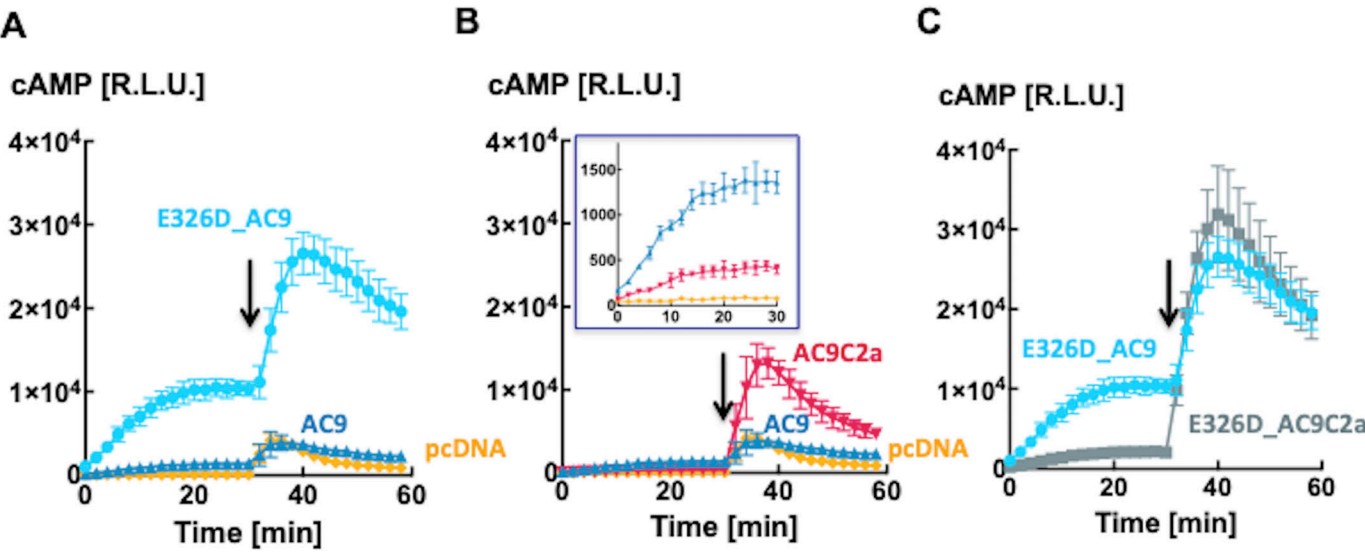

**Figure 2. Levels of cAMP in HEK293FT cells expressing adenylyl cyclase 9 variants.**
**(A)** Mutation E326D markedly enhances cellular levels of cAMP. Intracellular levels of cAMP reported by GloSensor22F firefly bioluminescence in HEK293FT cells transiently transfected with AC9 (triangles, ▲), E326D_AC9 (circles, ●), or the skeleton vector pcDNA3 (diamonds, ♦). Isoproterenol (10 nM) was applied as indicated by the arrow. Data are the mean ± S.D. n = 4/group, representative of three independent experiments. **(A, B)** Effects of removing the isoform-specific C2b domain on intracellular levels of cAMP reported by GloSensor22F firefly bioluminescence from (A) HEK293FT cells transiently transfected with AC9 (triangles, ▲), AC9C2a (wedges, ▼), or pcDNA3 (diamonds, ♦). The insert shows the basal cAMP levels from the same wells. **(A)** Traces of AC9 and pcDNA3 are the ones already shown in (A). **(C)** HEK293FT cells transiently transfected with E326D_AC9 (squares, ■) or E326D_AC9C2a (circles, ●). **(A)** Traces of E326D_AC9 are the ones already shown in (A). Isoproterenol (10 nM) was applied as indicated by the arrow. Data are the mean ± S.D., n = 4/group, representative of three independent experiments. Source data are available for this figure.

inhibition by C2b was reduced by the E326D mutation (Fig 2C). The statistical analysis of this experiment is shown in Fig 3. Closely similar results were obtained with prostaglandin E1 as the agonist (Fig S1), and further iterations with isoproterenol are provided as Figs S2 and S3, to illustrate the reproducibility of the outcome between different batches of transfected cells.

The lack of a substantial enhancement of the response to isoproterenol in E326D_AC9C2a is unlikely to be due to the saturation of GloSensor22F as it was apparent with all agonist-induced responses that evoked light emission well below those elicited by the quality control forskolin/rolipram stimulus. Comparison of the expression of the AC9 proteins examined here showed that AC9 and AC9C2a were consistently present at higher levels than their E326D counterparts (Fig 4). Thus, the cAMP-producing capacities of the E326D mutants are likely underestimated when compared to the WT variants. Importantly, the levels of expression of the AC9 proteins lacking C2b were similar to the respective full-length versions (Fig 4). As a quality control for GloSensor22F expression was run in each well, and as GloSensor22F is validated for the scalar analysis of intracellular cAMP levels (Binkowski et al, 2011; Felouzis et al, 2016; Goulding et al, 2018; Baldwin et al, 2019; Ayukawa et al, 2020; Hoy et al, 2020), it is justified to conclude that the dramatic differences of cellular cAMP levels observed in our experiments largely reflect the respective rates of cAMP production.

The data show that in addition to quenching the response to GsR, the C2b domain has an auto-stimulatory effect on AC9. Moreover, perturbation of the coiled-coil of AC9 can differentially modify the influence of the C2b domain, in that it sustains auto-stimulation but appears to suppress auto-inhibition. Conformational changes of the coiled-coil could occur under physiological conditions. Although

full-length AC9 is resistant to regulation by G proteins, the interfaced helices 6 and 12 of the transmembrane arrays continue into the cytoplasm as the transducing helices and form the coiled-coil (Qi et al, 2019). The transmembrane arrays may function as cell-surface receptors and/or sensors of the lipid composition of the plasma membrane, and changes in the relative positions of helices 6 and 12 may be transmitted to the coiled-coil (Finkbeiner et al, 2019; Qi et al., 2019, 2022; Seth et al, 2020). Overall, this scenario appears similar to the activation of soluble guanylyl cyclase by nitric oxide (Horst et al, 2019).

Basal cAMP production by AC9 is inhibited by an intracellular pathway involving $Ca^{2+}$ and calcineurin (Antoni et al, 1995, 1998; Paterson et al, 1995; Cumbay & Watts, 2005). Hence, it is part of regulated intracellular signalling circuits. In contrast to our results in intact cells, purified preparations of bovine AC9 and AC9C2a (Qi et al, 2019) showed no difference in basal enzymatic activity. This could be due to the use of high, activity-stimulating concentrations of $Mn^{2+}$ (5 mM) in the cyclase assay (Dessauer et al, 2002; Qi et al, 2019). Alternatively, as the C2b domain may be phosphorylated (10 documented sites) and ubiquitinated (three sites) (Antoni, 2020), it is possible that post-translational modifications are essential for the auto-stimulation observed in HEK293FT cells, and these are likely to have been lost during the multi-step purification process (Qi et al, 2019). Indeed, functionally relevant activation of AC9 by protein kinase cascades independently of Gs has been reported in neutrophil granulocytes (Liu et al, 2010). A yet further possibility is that auto-stimulation may require additional protein(s) lost during purification.

The auto-inhibitory motif of C2b that occludes AC9 when in complex with Gsα is well delineated (Pálvölgyi et al, 2018; Qi et al, 2019). With respect to how C2b might stimulate AC9 activity, indirect evidence points to the forskolin binding pocket (Tang & Hurley,

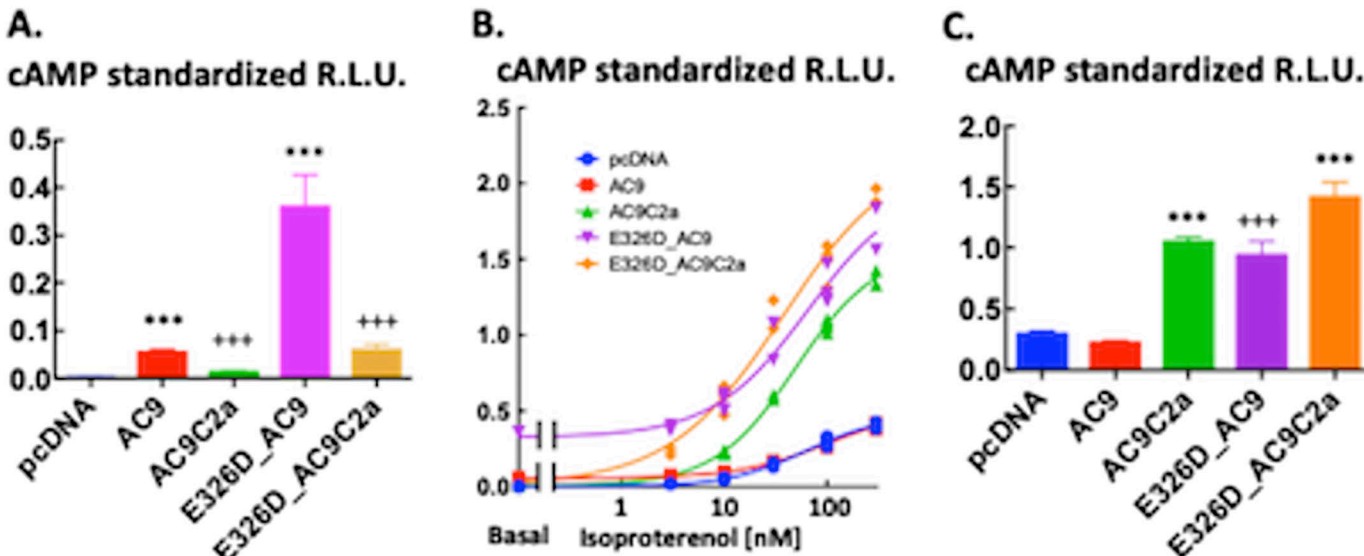

**Figure 3. Auto-stimulation of basal and auto-inhibition of isoproterenol-evoked cAMP production by the C2b domain of AC9.**
Statistical analyses of the data are presented in Fig 2. **(A)** Basal intracellular cAMP levels reported by GloSensor22F bioluminescence in HEK293FT cells transiently expressing pcDNA3, AC9, AC9C2a, E326D_AC9, or E326D_AC9C2a. Data are the average of the apparent plateau reached at 25–30 min of incubation (see Fig 2 for time courses) standardized in each well by the peak of the response elicited by 5 $\mu$M forskolin and 100 $\mu$M rolipram administered at the end of the experiment. Data are means ± S.D. n = 14/group. As the variances of the five groups were statistically different, the data analysis was carried out after log transformation of the data. One-way ANOVA, $F_{(4,65)}$ = 1,464, $P < 0.0001$; Tukey's post hoc multiple comparison test: ***$P < 0.001$ versus respective variant lacking the C2b domain, and +++$P < 0.0001$ versus the pcDNA3 group. **(A, B)** Concentration–response to isoproterenol from the cells described in (A). All measurements of the standardized peak responses are shown. Curves were fitted by non-linear four-parameter regression with a variable slope in GraphPad Prism v.6. The bottom of the curve was set as the respective basal values. There was no difference between the EC50 values at $\alpha$ = 0.05; the range was 37–63 nM. The maximal responses were E326D_AC9C2a = E326D_AC9 = AC9C2a > AC9 = pcDNA3, where > denotes statistical significance at $P < 0.05$, as indicated by the 95% confidence intervals calculated by the non-linear regression algorithm. **(C)** Increment over the respective basal levels induced by 100 nM isoproterenol calculated by averaging three consecutive time-points once the peak level of bioluminescence was reached (see Fig 2 for time courses). The relative light units were standardized in each well by the peak of the response elicited by 5 $\mu$M forskolin and 100 $\mu$M rolipram administered at the end of the experiment. Means ± S.D. n = 4/group. One-way ANOVA, $F_{(4,15)}$ = 169.9, $P < 0.0001$; Tukey's post hoc multiple comparison test: ***$P < 0.0001$ versus respective full-length variant, and +++$P < 0.0001$ versus the pcDNA3 group. The results for 10 nM isoproterenol were closely similar.
Source data are available for this figure.

1998; Qi et al, 2022). First, the cryo-electron microscopic map of bovine AC9 shows that the C2b domain is capable of short-distance interactions with residues in the forskolin binding pocket (Qi et al, 2019). Second, database analysis of the phylogenetic development of AC9 reveals that a long (>100 amino acid residues) C2b domain containing the highly conserved auto-inhibitory motif (Pálvölgyi et al, 2018) only features in vertebrate AC9-s. Simultaneously, a well-defined, "low-reactivity to forskolin" configuration (Tang & Hurley, 1998; Yan et al, 1998) of the C2a catalytic domain also emerges. These features are already present in lamprey and hagfish AC9, the two earliest vertebrate species alive today. In contrast, the invertebrate homologs of AC9, including those of the chordate (Amphioxiformes) species, have short C2b domains and their C2a domains are in the "high-reactivity to forskolin" configuration. Hence, it seems reasonable to suggest that the "low-reactivity to forskolin" configuration of the C2a domain of AC9 is instrumental to auto-stimulation by the C2b domain.

## Final summary

Fig 5 shows the three modes of operation of AC9 suggested by this study. In mode 1, basal activity is driven by the C2b domain associating with the forskolin binding pocket, whereas the coiled-coil exerts an inhibitory effect. In the presence of active Gs$\alpha$, the C2b domain is displaced into the active site of AC9 (Qi et al, 2019), thus

preventing activation by Gs$\alpha$ (Pálvölgyi et al, 2018). This mode of operation appears geared to support constitutively active cAMP-dependent processes. Amongst others, the maintenance of re-leasable pools of secretory vesicles (Nagy et al, 2004), the activity of ion channels (Antos et al, 2001; Baldwin et al, 2021), and vesicle trafficking in the trans-Golgi network require such input (Muniz et al, 1997). Indeed, AC9 has been specifically implicated in the latter process (Cancino et al, 2014). In mode 2, destabilization of the coiled-coil results in high basal levels dependent on C2b and appears to reduce the efficacy of auto-inhibition amounting to a paradigm switch. In our studies, a human mutation induced this mode of operation. However, it is entirely conceivable that lipid mediators could destabilize the coiled-coil by changing the relative positions of transmembrane helices 6 and 12 (Bassler et al, 2018; Qi et al, 2019). In mode 3, when the C2b domain is removed, AC9 has low basal activity and is robustly responsive to activation by GsCR, a further paradigm switch. On the basis of current knowledge, in mammals this change would require proteolytic cleavage (Antoni, 2020) and is apparent in heart tissue (Pálvölgyi et al, 2018). Interestingly, several teleost species have a second AC9 gene featuring a long C2b domain that lacks the auto-inhibitory motif (Antoni, 2020), indicating that the dichotomy of separate GsCR responsive and unresponsive AC9 species has adaptive significance. In principle, all three modes of operation may occur simultaneously in the same cell. As AC9 is widely distributed in the

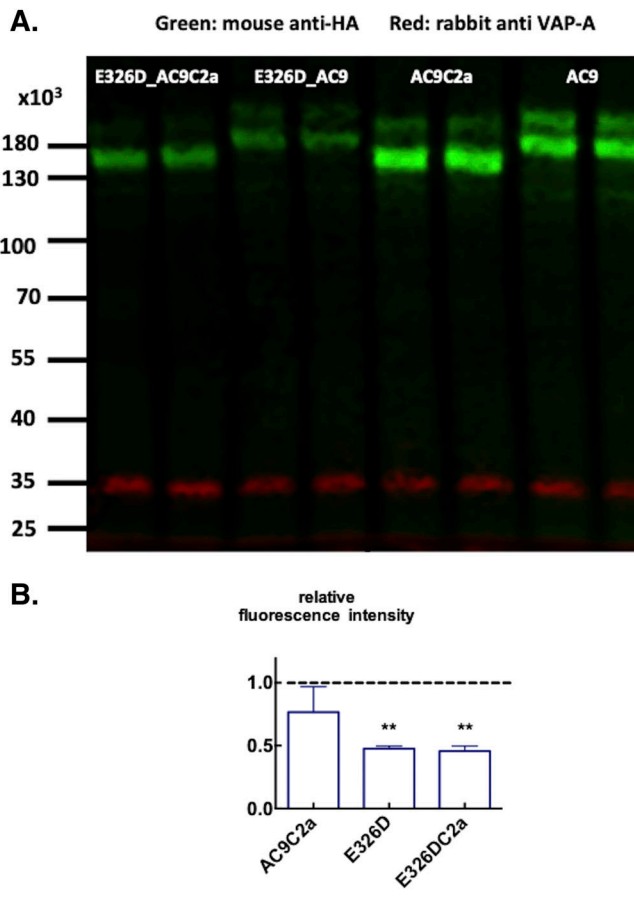

**A.**

Green: mouse anti-HA    Red: rabbit anti VAP-A

**B.**

relative
fluorescence intensity

**Figure 4. Detection of the expression of adenylyl cyclase 9 variants by immunoblots.**
**(A)** Immunoblot of extracts of crude membranes prepared from HEK293FT cells transfected with (from left to right, two lanes each) E326D_AC9C2a, E326D_AC9, AC9C2a, AC9. Green bands show reaction with anti_HA; red bands are stained for VAMP-associated protein 33 (VAP-A). The relative intensities of the protein bands reported by Empiria software (LI-COR Biosciences) were E326D_AC9C2a 0.42, E326D_AC9 0.40, AC9C2a 1.1, and AC9 1.0; the blot is representative of three transfections. The numbers on the left indicate the migration of the molecular size markers on the gel. **(B)** Relative fluorescence intensities of anti-HA immunoreactive bands in three separate transfections. Mean ± S.D., n = 3/ group. **$P$ < 0.01 significantly different from unity by a one-sample $t$ test.

body, these features of intracellular cAMP signalling are bound to be relevant in several organ systems.

## Materials and Methods

### cDNA constructs

Human AC9 tagged N-terminally with haemagglutinin antigen (HA) and C-terminally with FLAG in pcDNA3.1 (AC9) was a gift of Dr. Adrienn Pálvölgyi (Egis PLC). This construct was used for site-directed mutagenesis with Phusion polymerase (New England Biolabs) following previously published protocols (Xia et al, 2015). Glutamate at position 326 is part of the interface of the coiled-coil of AC9 (Fig 2B and C). Subsequently, the C2b domain was removed

from AC9 and E326D_AC9 (Pálvölgyi et al, 2018) to encode AC9_C2a and E326D_AC9C2a, respectively. The sites of the mutations and the N- and C-terminal coding sequences were verified by Big Dye sequencing.

### Cells

Fast-growing human embryonic kidney 293T cells (HEK293FT) were maintained in DMEM/10% FBS (vol/vol) and passaged at 5- to 7-d intervals with TrypLE Express (Invitrogen/Gibco) to detach the cells. About 2 million cells in 1.6 ml of growth medium were mixed with 4 $\mu$g of AC9 cDNA or 2 $\mu$g of pcDNA3 plus 2 $\mu$g of GloSensor22F (Binkowski et al, 2011) pre-complexed with Lipofectamine 2000 in 400 $\mu$l Opti-MEM. GloSensor22F encodes a firefly luciferase–based biosensor that, when provided with its substrate luciferin, emits light in proportion to the amount of cAMP bound to it (Binkowski et al, 2011).

### Measurement of cellular cAMP levels

After 48 h, the transfected cells were plated in poly-L-lysine–coated, 96-well white tissue culture plates (Greiner) at $10^5$ cells per well and incubated as above for 24 h. Subsequently, the cells were depleted of serum in DMEM for 60 min and incubated in Hank's balanced salt solution containing 1 mM $MgSO_4$, 1.5 mM $CaCl_2$, 10 mM Hepes, pH 7.4, and 1 mM beetle luciferin (Promega) at 32°C for a further 60 min. The plates were transferred to a BMG LUMIstar Ω plate reader, and the luminescence signal was recorded at 32°C from each well at 2-min intervals. Usually, a further 30 min was required for the basal light signal to stabilize. Drug treatments were added from a 12-channel hand-held pipette. Because of the in-herent variability of transient transfections, each well received a mixture of 5 $\mu$M forskolin (LC Labs) and 100 $\mu$M rolipram (Insight Biotech) at the end of the recording as a quality control stimulus. This standardization was possible because AC9 is not stimulated by 5 $\mu$M forskolin even when stimulated by Gs$\alpha$ (Baldwin et al, 2019; Qi et al, 2019, 2022). As the GloSensor response at high levels of cAMP becomes non-linear and eventually saturates (Binkowski et al, 2011), the concentration–response curves may be flattened at high concentrations of isoproterenol. This is likely to be the case when the isoproterenol response (largely gen-erated by the transfected AC9 variant) is higher than the stan-dardizing stimulus that is largely produced by the host cell adenylyl cyclases.

### Immunoblots

Expression of HA/FLAG-tagged AC9 proteins was examined by SDS–PAGE and immunoblotting of extracts prepared from crude membranes of transfected HEK293FT cells as reported previously (Antoni et al, 1998; Pálvölgyi et al, 2018). Protein blots were reacted with the 12CA5 anti-HA (Abcam) or M2 anti-FLAG mouse mono-clonal antibodies (Sigma-Aldrich) in conjunction with rabbit anti-VAP-A (gift of Dr Paul Skehel) (Skehel et al, 2000) as a sample loading control marker. Secondary IRDye 680RD-tagged anti-mouse and IRDye 800CW-tagged anti-rabbit goat IgGs were from LI-COR Biosciences with fluorescence read-out in a LI-COR

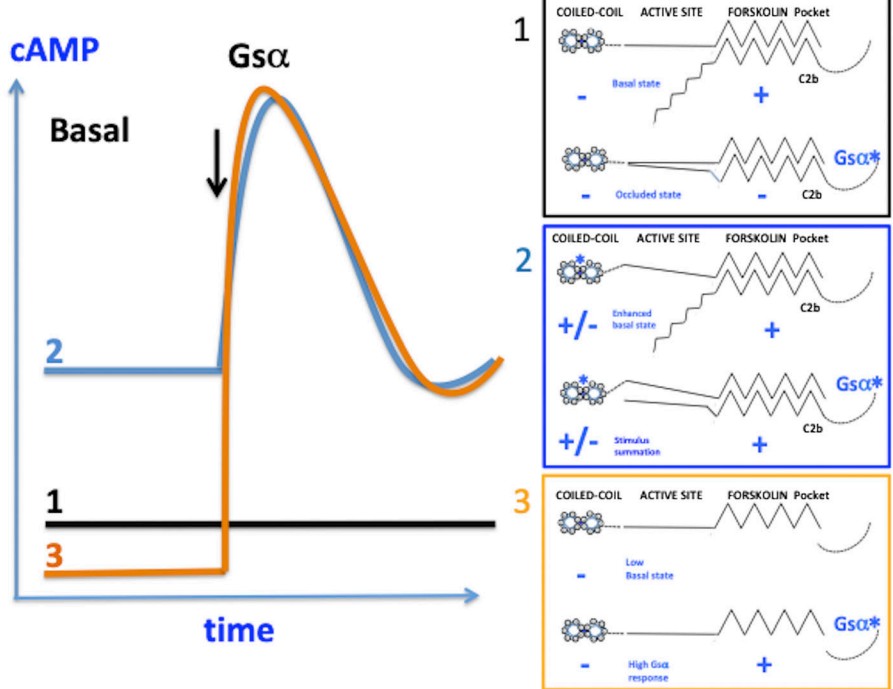

**Figure 5. Three modes of operation of AC9 as indicated by this study and previous work.**
Mode 1: high basal activity supported by the C2b domain and quenching of activation "occluded state" in the presence of activated Gsα, as shown in the previous work. Importantly, the production of cAMP by AC9 is inhibited by intracellular free $Ca^{2+}$ through calcineurin. Mode 2: conformational change at the coiled-coil (E326D mutation in this study) enhances basal activity dependent on C2b and relieves the quenching of the activation by Gsα. Thus, the enzyme may summate input to the coiled-coil and the stimulation by Gsα. Mode 3: removal of the C2b domain markedly reduces basal activity and enables stimulation by Gsα; that is, AC9 now resembles a conventional adenylyl cyclase stimulated by Gsα.

Odyssey imager. Only HA-tag staining was used for quantification as the staining with anti-FLAG M2 antibody appeared to show context dependence.

## Database searches

The NCBI GenBank and the Wellcome Trust Ensembl Genome Browser servers were used to find AC9-related sequences by BLAST searches. Protein sequence alignments were carried out with the Clustal Ω web application on the European Bioinformatics Institute server (Madeira et al, 2022). The cryo-electron microscopic maps of bovine AC9 were downloaded from the RCSB Protein Data Bank server.

## Supplementary Information

## Acknowledgements

This study was funded by the University of Edinburgh where Z Chen was a student in the MScRes course in Biomedical Sciences. We would like to thank Miss Heather McClafferty for help with cell lines and various reagents; Dr Paul LeTissier for access to a luminometer; Prof Mike Shipston, Dr Paul Skehel, Dr Sutherland MacIver, and Dr Tamás Balla for helpful discussions; and Dr Mandy Jackson and Dr Paul Skehel for access to laboratory facilities and reagents.

## Author Contributions

Z Chen: data curation, formal analysis, and investigation.
FA Antoni: conceptualization, data curation, formal analysis, supervision, investigation, methodology, and writing—original draft, review, and editing.

## Conflict of Interest Statement

The authors declare that they have no conflict of interest.

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
