## [Reviewer comments · Life Science Alliance]

Life Science Alliance

Adenylyl cyclase 9 is auto-stimulated by its isoform-specific C-terminal domain

Zhihao Chen and Ferenc Antoni

DOI: <https://doi.org/10.26508/lsa.202201791>

Corresponding author(s): *Ferenc Antoni, University of Edinburgh*

Review Timeline:

Submission Date:	2022-10-31
Editorial Decision:	2022-12-03
Revision Received:	2022-12-21
Editorial Decision:	2023-01-03
Revision Received:	2023-01-06
Accepted:	2023-01-09

Scientific Editor: Novella Guidi

Transaction Report:

December 3, 2022

Re: Life Science Alliance manuscript #LSA-2022-01791-T

Dr. Ferenc Antoni
University of Edinburgh
United Kingdom

Dear Dr. Antoni,

Thank you for submitting your manuscript entitled "Human adenylyl cyclase 9 is auto-stimulated by its isoform-specific C-terminal domain and equipped with paradigm-switching mechanisms" to Life Science Alliance. The manuscript was assessed by expert reviewers, whose comments are appended to this letter. We invite you to submit a revised manuscript addressing the Reviewer comments.

Thank you for this interesting contribution to Life Science Alliance. We are looking forward to receiving your revised manuscript.

Sincerely,

B. MANUSCRIPT ORGANIZATION AND FORMATTING:

Reviewer #1 (Comments to the Authors (Required)):

The manuscript by Chen and Antoni examines the role of the C-terminal portion of AC9 in the regulation of the enzyme. This AC isoform is understudied and clearly unique among transmembrane ACs in the way it is regulated by GPCRs and forskolin. The present work breaks new ground in understanding the complex mechanisms of AC9 function and regulation and the results are highly novel and important. The work would be of high interest to the readers of the Journal.

Specific comments:

1. The data presentation in Figure 2 does not represent the variability in the data for the reader. Instead of showing two representative plots it would be preferred to plot the mean and SD of all the experiments using error bars. Statistical analyses appear appropriate but the representation of the data in these figures does not properly convey this to the reader.
2. Figure 4 shows differences in the level of expression of the different constructs. Since the authors are making this point then there should be a graph added showing the reproducibility of the data by plotting the mean {plus minus} SD of the multiple immunoblots that were done (in addition to the one representative blot).
3. Is Figure 3B plotted on a log scale?

Reviewer #2 (Comments to the Authors (Required)):

The manuscript by Chen and Antoni describes an interesting new mode of AC9 regulation by the C-terminal C2b domain, whereby GPCR coupling is dependent on the C2b. The mutation E326D stimulates AC9, increasing the basal activity. This is also dependent on the C2b domain. The authors conclude that C2b controls the basal activity of AC9 and reduces the ability of GPCRs to activate the enzyme. The interplay of the signalling helices (coiled coil) and the C2b domain can thus regulate the signaling inputs via receptors.

The authors propose an interesting mechanism of AC regulation, which will be of value for scientists in this field. Overall the text is well written, although the quality and clarity of the text and figures can still be improved. On the experimental side, one aspect that is prominently missing in the manuscript is the analysis of AC9 mutant localisation in the cells. This may be particularly important for the GPCR coupling, which the authors probe using receptor ligands. Although the authors show that the expression levels of the mutated constructs consistent with the observed differences in activity, the trafficking of the mutated AC9 variants has not been explored. The authors should consider performing any of the experiments that can help understand better the status of the mutated AC9 in the cells. This could be done using any number of techniques, such as immunocytochemistry, fluorescence microscopy, cell surface biotinylation, endoH resistance and so on. The arguments will be strengthened by a more comprehensive analysis of the expressed proteins.

What is AC9C2a? It is prominently featured in the figures and in the text, introduced first on top of page 5. But the authors do not explain what exactly this is. Is it the C2a domain? Or is it a construct lacking the C2b domain (presumably this is the case)? The authors would help the readers appreciate the study by removing the need to decipher what is what in the manuscript - in this case a more suggestive name for the construct might be useful.

Page 7 - bottom reference is surrounded by square symbols.

Fig. 2 and other curve figures. To improve the graphs, the authors could consider using a uniform style for all data points (e.g., circles). Since they have coloured the curves and the datapoints, changing shapes becomes redundant and adds unnecessary complexity to the graphs. This is a minor suggestion to improve the style of presentation, not to the substance of the graphs.

Fig. 4 - please add some markers on the left or right side of the gel, so that we know where to look and which bands are of interest.

There is no reason why Fig. 4 should not be merged with Figure 2, where the AC9C2a construct is first introduced (for example as Figure 2D).

Figure 5 and the corresponding discussion - the authors should probably discuss this suggestion in greater detail. The current description of their idea is minimal. Some illustration depicting this idea could be added to this figure, to not only show and-drawn curves (which is a rather abstract representation of what is happening), but to also represent visually the authors' proposal for AC9 regulation by the C2b.

Figs. S2 and S3 - these two figures are confusing. Are we looking at the same or very similar data? Why do these figures need to be split into two?

Fig. S2 and S3 would benefit from more descriptive titles (and if the two figures show the same thing, they should be merged).

Reviewer #1 (Comments to the Authors (Required)):

The manuscript by Chen and Antoni examines the role of the C-terminal portion of AC9 in the regulation of the enzyme. This AC isoform is understudied and clearly unique among transmembrane ACs in the way it is regulated by GPCRs and forskolin. The present work breaks new ground in understanding the complex mechanisms of AC9 function and regulation and the results are highly novel and important. The work would be of high interest to the readers of the Journal.

We would like to thank Reviewer 1 for the favourable comments on the manuscript.

Specific comments:

1. The data presentation in Figure 2 does not represent the variability in the data for the reader. Instead of showing two representative plots it would be preferred to plot the mean and SD of all the experiments using error bars. Statistical analyses appear appropriate but the representation of the data in these figures does not properly convey this to the reader.

A revised figure is provided with mean \pm SD, of an experiment carried out on a single batch of transfected cells in quadruplicate. Further analogous experiments from different batches of transfected cells are shown in the Supplementary figures. The prime purpose of these figures is to demonstrate the reproducibility of the findings between different transfections.

2. Figure 4 shows differences in the level of expression of the different constructs. Since the authors are making this point then there should be a graph added showing the reproducibility of the data by plotting the mean {plus minus} SD of the multiple immunoblots that were done (in addition to the one representative blot).

The band intensities from three different transfections are shown as Fig 4B.

3. Is Figure 3B plotted on a log scale?

Indeed, all concentration-response curves are shown on a log₁₀ scale as per convention in pharmacologic studies. An axis break was shown between 0 (basal) and 3 nM as the Log₁₀ of zero is not defined. This was done to

indicate that the bottom values for curve fitting by non-linear regression were set as the respective basal levels. In order to avoid confusion we have changed "0" to "Basal" and provide a corresponding explanation in the figure legends.

Reviewer #2 (Comments to the Authors (Required)):

The manuscript by Chen and Antoni describes an interesting new mode of AC9 regulation by the C-terminal C2b domain, whereby GPCR coupling is dependent on the C2b. The mutation E326D stimulates AC9, increasing the basal activity. This is also dependent on the C2b domain. The authors conclude that C2b controls the basal activity of AC9 and reduces the ability of GPCRs to activate the enzyme. The interplay of the signalling helices (coiled coil) and the C2b domain can thus regulate the signaling inputs via receptors.

The authors propose an interesting mechanism of AC regulation, which will be of value for scientists in this field. Overall the text is well written, although the quality and clarity of the text and figures can still be improved. We would like to thank Reviewer 2 for the largely positive evaluation of the manuscript.

On the experimental side, one aspect that is prominently missing in the manuscript is the analysis of AC9 mutant localisation in the cells. This may be particularly important for the GPCR coupling, which the authors probe using receptor ligands. Although the authors show that the expression levels of the mutated constructs consistent with the observed differences in activity, the trafficking of the mutated AC9 variants has not been explored. The authors should consider performing any of the experiments that can help understand better the status of the mutated AC9 in the cells. This could be done using any number of techniques, such as immunocytochemistry, fluorescence microscopy, cell surface biotinylation, endoH resistance and so on. The arguments will be strengthened by a more comprehensive analysis of the expressed proteins.

We fully agree that trafficking and intracellular localization of adenylyl cyclases is important for the understanding the biological role(s) of these proteins e.g. see F Antoni (2006) DOI 10.1007/s11064-005-9019-1. Unfortunately, in our experience, transient transfections of HEK293FT cells are not very

informative in this respect. Upon immunostaining, a spectrum of morphologies is observed, ranging from largely plasma membrane-like distribution in flat cells with immunolabelled processes, to rounded cells showing intensive staining throughout the cytoplasm. Various intermediate patterns between these two extremes are apparent. In HEK293 cells stably overexpressing AC9 or AC9C2a, the membrane-like distribution predominates by far for both proteins (Paterson et al JNeurochem, 2000, Pálvölgyi et al CellSignal, 2018). The structural basis of autoinhibition by the C2b domain is well delineated (Pálvölgyi et al CellSignal, 2018, Qi et al Science 2019) and does not seem to be the result of differential intracellular trafficking.

The primary aim of our paper is to communicate the novel and remarkable regulatory repertory of AC9, including the properties of a human mutant. The molecular mechanisms underlying these functionally distinct phenotypes require longer-term, specialized analyses. We are confident that the data we communicate here are novel, highly reproducible and suitable as the starting point of in-depth mechanistic analyses including structural biology and intracellular trafficking.

What is AC9C2a? It is prominently featured in the figures and in the text, introduced first on top of page 5. But the authors do not explain what exactly this is. Is it the C2a domain? Or is it a construct lacking the C2b domain (presumably this is the case)? The authors would help the readers appreciate the study by removing the need to decipher what is what in the manuscript - in this case a more suggestive name for the construct might be useful.

Thank you for highlighting – this is one of the problems of putting Materials and Methods at the end of a communication. AC9C2a as well as E326D_AC9C2a were introduced and defined at the bottom of paragraph 3 on page 8 of the original manuscript. Indeed, both indicate the removal of the C2b domain. We have reworded the manuscript so that the definitions are clear in the Results.

Page 7 - bottom reference is surrounded by square symbols.

Thank you for noticing. These squares seem to be inserted by the online .pdf conversion by the journal. They are not found if the .docx file is saved as .pdf on the local computer.

Fig. 2 and other curve figures. To improve the graphs, the authors could consider using a uniform style for all data points (e.g., circles). Since they have coloured the curves and the datapoints, changing shapes becomes redundant and adds unnecessary complexity to the graphs. This is a minor suggestion to improve the style of presentation, not to the substance of the graphs.

Thank you, the purpose of this was to make sure that assuming a printed version is in not published/printed in colour, readers could discern the curves. As per the request by Reviewer 1 we have changed the graphs to show means \pm S.D. for each time-point, which made the coding of the curves simpler.

Fig. 4 - please add some markers on the left or right side of the gel, so that we know where to look and which bands are of interest.

There is no reason why Fig. 4 should not be merged with Figure 2, where the AC9C2a construct is first introduced (for example as Figure 2D).

Markers were present in the original figure, one can only speculate that the pdf conversion gremlin left them off.

As per a request from Reviewer 1 we now show the average relative fluorescence intensities of the observed immunoreactive bands from three different batches of transfected cells hence the figure is bigger.

Figure 5 and the corresponding discussion - the authors should probably discuss this suggestion in greater detail. The current description of their idea is minimal. Some illustration depicting this idea could be added to this figure, to not only show and-drawn curves (which is a rather abstract representation of what is happening), but to also represent visually the authors' proposal for AC9 regulation by the C2b.

We have tried to improve the discussion and provided schematic cartoons for each functional state of the enzyme as requested. We also provide commentary as to where each of the modes of operation of AC9 may be relevant.

Figs. S2 and S3 - these two figures are confusing. Are we looking at the same or very similar data? Why do these figures need to be split into two? Fig. S2 and S3 would benefit from more descriptive titles (and if the two figures show the same thing, they should be merged).

These figures serve to demonstrate the reproducibility of the data between different batches of transfected cells as stated in the first sentence of the Appendix. Given the potential variability of transient transfections, it is justified to present these data separately, in order to show that each independent experiment gave closely similar results.

January 3, 2023

RE: Life Science Alliance Manuscript #LSA-2022-01791-TR

Dr. Ferenc Antoni
University of Edinburgh
Discovery Brain Sciences
15 George Sq
Edinburgh EH8 9XD
United Kingdom

Dear Dr. Antoni,

Thank you for submitting your revised manuscript entitled "Adenylyl cyclase 9 is auto-stimulated by its isoform-specific C-terminal domain". We would be happy to publish your paper in Life Science Alliance pending final revisions necessary to meet our formatting guidelines.

- please upload your supplementary figures as single files and add the supplementary figure legends to the main manuscript text
- please add the Twitter handle of your host institute/organization as well as your own or/and one of the authors in our system
- please add the author contributions and a conflict of interest statement to the main manuscript text

A. FINAL FILES:

B. MANUSCRIPT ORGANIZATION AND FORMATTING:

Sincerely,

January 9, 2023

RE: Life Science Alliance Manuscript #LSA-2022-01791-TRR

Dr. Ferenc Antoni
University of Edinburgh
Discovery Brain Sciences
15 George Sq
Edinburgh EH8 9XD
United Kingdom

Dear Dr. Antoni,

Thank you for submitting your Research Article entitled "Adenylyl cyclase 9 is auto-stimulated by its isoform-specific C-terminal domain". It is a pleasure to let you know that your manuscript is now accepted for publication in Life Science Alliance. Congratulations on this interesting work.

DISTRIBUTION OF MATERIALS:

Again, congratulations on a very nice paper. I hope you found the review process to be constructive and are pleased with how the manuscript was handled editorially. We look forward to future exciting submissions from your lab.

Sincerely,
